# Ultrasound classification of non-mass breast lesions following BI-RADS presents high positive predictive value

Mingnan Lin, Size Wu *

Department of Ultrasound, The First Affiliated Hospital of Hainan Medical University, Haikou, Hainan Province, People's Republic of China

* wsz074@aliyun.com

## Abstract

### Purpose

To investigate the positive predictive value of ultrasound classification of non-mass breast lesions (NMLs) following breast imaging reporting and data system (BI-RADS), and enhance understanding of NMLs.

### Materials and methods

Fifty-nine women with 59 ultrasound-detected breast NMLs were finally enrolled. The ultrasound (US) features of breast NMLs were analyzed; the incidence of malignant NMLs was calculated; the malignancy risk stratification of US for breast NMLs was established using BI-RADS.

### Results

The incidence of malignant NMLs was 4.59% of all breast carcinoma. Non-ductal hypoechoic area, microcalcifications and posterior shadowing are the main US features of malignant NMLs, and there were significant differences between malignant and benign NMLs for microcalcifications and posterior shadowing. Taking BI-RADS 4B as a cutoff value, the sensitivity, specificity, area under the receiver operating characteristic curve (AUC), positive and negative predictive values, and odds ratio of the BI-RADS category were 82.98%,41.67%,0.62,84.78%,38.46% and 3.48, respectively.

### Conclusions

Stratifying the malignancy risk of breast NMLs using the BI-RADS the sensitivity and positive and predictive value are promising, but the likelihood of malignancy of malignant NMLs is underestimated, and that of benign NMLs is overestimated. The solution may be that to separate NMLs from breast masses and use different malignancy risk stratification protocols.

**Data Availability Statement:** All data are present in the paper and its Supporting Information files.

**Funding:** This study was financially supported by the National Natural Science Foundation of China (Grant No. 81560290).

**Competing interests:** The authors have declared that no competing interests exist.

# Introduction

Breast lesions are common in the adult female population. While some lesions are palpable, and the others may be vague or occult. Currently, breast lesions can be detected and assessed using mammography, ultrasound and magnetic resonance imaging (MRI). Mass is a common term used for findings in various imaging modalities in the Breast Imaging Reporting and Data System (BI-RADS) descriptive lexicon, in which the mass is defined as a three-dimensional space-occupying lesion that is observed on two different projections and can be distinguished from normal anatomic structures [1–3]. Not all breast lesions manifest features meeting the exact BI-RADS US criteria in clinical practice. There are some non-mass lesions (NMLs) or non-mass-like lesions on US images, which causes an issue for the depiction of breast lesions using the standard lexicon. There is currently no standardized definition, terminology or guidelines for interpretation of a NML on US image, which affects interpretative accuracy and management.

Breast cancer is a round, oval or irregular shaped entity, with or without extra associated pathological lesions, manifesting as a mass or NML-like appearance on medical images. Some breast benign lesions mimic malignant lesions in appearance. There is considerable overlap between the US features of benign and malignant breast NMLs. In previous routine clinical practice, except a few of NMLs were classified as BI-RADS category 0, other NMLs were actually classified as a certain category. This results in an inappropriate malignancy risk stratification of breast masses and NMLs. Therefore, one or more reliable and non-invasive US findings that could reduce the number of unnecessary procedures would be valuable. There are several descriptions of breast NMLs on US, but none of them have been adopted in consensus [2, 3]. To date, knowledge and understanding about breast NMLs on US are still insufficient. Previous study showed that US BI-RADS exhibits high efficacy for the assessment of breast masses, with sensitivity of 92%, specificity of 85% and accuracy of 87% for BIRADS 3–5 lesions [4]. However, there is no protocol for the malignancy risk assessment of breast NMLs. The purpose of this study was to investigate the positive predictive value and other performances of malignancy risk stratification of the breast NMLs on US images following BI-RADS in a try, and enhance understanding of NMLs.

# Materials and methods

## Study population

In this cross-sectional retrospective study, consecutive patients who had undergone breast surgery, coarse needle biopsy, and an US-guided vacuum-assisted biopsy between January 2017 and December 2021 in a tertiary hospital were retrieved from the informatics database to comprise the target population. We retrieved all adult patients with histology of the breast, correlated them with the informatics database for the US examination and patients' history, and reviewed the US images. The inclusion criteria were that the patients who had undergone US evaluation immediately before the above procedures, the quality of US images was good, and the images of the breast lesions met the features of breast NML on US addressed in the literature by Choe et al. that an area with or without an associated mass, accompanying calcifications, and architectural distortion and duct ectasia [2]. The exclusion criteria were that the patients with breast mass and NMLs did not have corresponding US images (undetected), male patients with breast mass lesions meeting the strict BI-RADS criteria of "mass" in consensus by two physicians-in-ultrasound (radiologists) (a junior with 2 years and a senior physician with 20 years of breast imaging experience), the breast lesions had a BI-RADS category 0 on US images, and the histopathological result of the breast lesion was indeterminant. If a patient

**Table 1. Baseline characteristics of patients with NMLs.**

| Characteristic | Malignant NML (n = 47) | Benign NML (n = 12) | *P* value |
|---|---|---|---|
| Age (y) | 51.11±10.27* | 42.42±10.59* | 0.12 |
| BMI (kg/m$^2$) | 23.36±2.73* | 23.25±2.34* | 0.90 |
| Family history of breast cancer | | | 1.00 |
| Yes | 1 | 0 | |
| No | 46 | 12 | |
| Breast pain | | | 1.00 |
| Yes | 18 | 5 | |
| No | 29 | 7 | |
| Nipple discharge | | | 1.00 |
| Yes | 0 | 0 | |
| No | 47 | 12 | |
| Palpable mass | | | |
| Yes | 0 | 0 | 1.00 |
| No | 47 | 12 | |
| Architectural changes | | | 1.00 |
| Skin thickening | 0 | 0 | |
| Architectural distortion | 0 | 0 | |
| No change | 47 | 12 | |
| Position of lesion | | | 1.00 |
| Left | 22 | 7 | |
| Right | 25 | 5 | |
| Quadrant | | | 0.53 |
| Upper exterior | 20 | 4 | |
| Upper interior | 7 | 3 | |
| Lower interior | 1 | 2 | |
| Lower exterior | 8 | 2 | |
| Enlarged lymph nodes in axillary regions | | | 1.00 |
| Yes | 2 | 0 | |
| No | 45 | 12 | |

Note: Unless otherwise indicated, data are numbers of NMLs and numbers in parentheses are percentages.

*Data are mean values, with standard deviation. BMI: Body mass index.

underwent two or more breast US, only the most recent US images and results were included. If a patient had two or more NMLs in a breast, only the malignant lesion or the most conspicuous representative benign lesions was included. If a patient underwent both biopsy and surgical excision, only the data from surgical excision were included. Based on these inclusion and exclusion criteria, 5,612 subjects with 5,795 lesions were recruited (4,671 benign lesions and 1,024 malignant lesions), and 5,553 subjects with 5,736 lesions were excluded, including 15 breast lesions (3 malignant lesion) miss detected by US, and 7 breast lesions (no malignant lesion) determined as BI-RADS category 0 by US. Finally, 59 women (mean age 43.3 years ± 10.8, age, range 30–78years) with 59 US-detected breast NMLs were included, for which final diagnoses were established, basing on histopathologic results of surgical excision (n = 52) and US-guided core needle biopsy (n = 7). Baseline characteristics of the 59 NMLs are summarized in **Table 1**.

A flowchart for inclusion of the study population is shown on **Fig 1**.

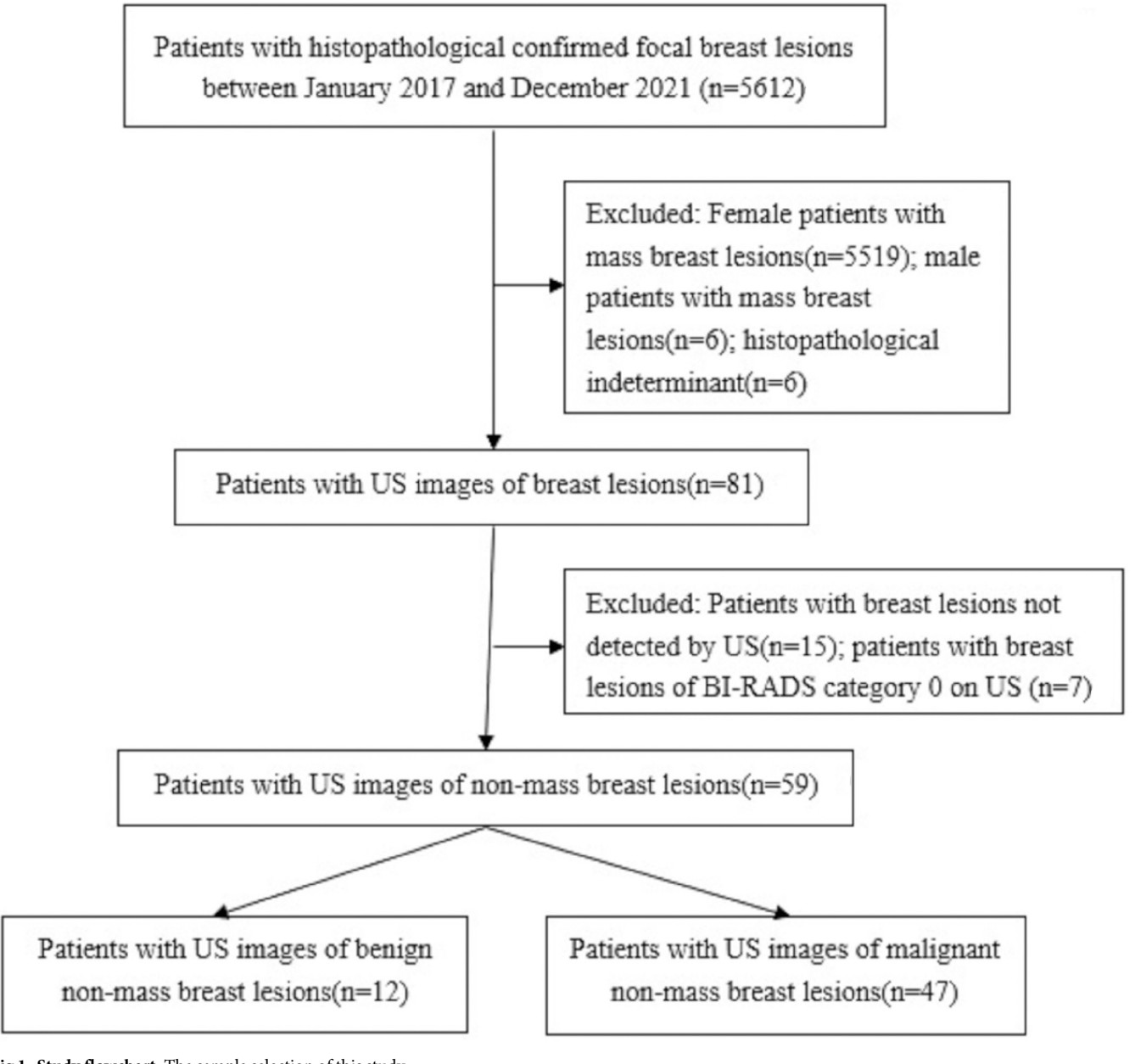

**Fig 1. Study flowchart.** The sample selection of this study.

## Ethics declaration

The data obtained from patients who had undergone US evaluation, biopsy and/or surgical operation, and all procedures followed were in accordance with the ethical standards of the responsible committee on human experimentation and with the World Medical Association Declaration of Helsinki (revised in 2000). The study was approved by the Ethics Committee of the first affiliated hospital of Hainan medical university [(2022, Scientific research (L) No.32)], and informed consent was waived because of the anonymized retrospective study design.

## Interpretation of the US images of NMLs

The US images of the breast NMLs were reviewed by two physicians-in-ultrasound (radiologists) together with reference to the definition, descriptions and associated US images for breast NMLs in the literature by Choe et al. [2]. Neither physicians-in-ultrasound knew the patients' clinical details or specific pathological results. The breast fatty tissue (isoechogenic) was used as the reference for comparison of echogenicity [1–3]. They stratified the malignancy risk of the breast NMLs in an analogous protocol that the BI-RADS for breast masses in consensus. Namely, in the assessment of NMLs, one or more descriptors (nonparallel orientation of the breast nodule, spiculated margins, angular margins, microcalcifications, or posterior shadowing) that commonly associated with histopathologically malignant nodules were used for the malignancy risk of NMLs [5].

## Acquisition of US images of the breast lesions

Breast US imaging had been performed for all patients referred to a tertiary hospital with suspicious breast lesions, breast pain, and for routine screening for malignancy. Multi-parameter US systems (Mindray DC 8; Mindray Resona 7; Aloka Prosound α-7; Aloka Prosound α-10; Siemens Acuson S2000; Phillips EPIQ5 and GE Logiq E9) were used for the breast examination by physicians-in-ultrasound with 5 to 30 years of experience in breast US. During the examination, the US systems were adjusted to small parts mode (breast), and a linear array transducer with a frequency of 5–15 MHz was used. The patient was instructed to take a supine position with the upper limb abduction, fully exposing the breast and axillary region. The breast was scanned carefully to detect any lesion. If a breast lesion or aberrant echogenic area was found, its location, shape, size, orientation (parallel and nonparallel), margin, echo pattern, posterior features, calcifications, architecture, vascularity, other associated findings, lymph node in the axillary region and other special findings were identified and scrutinized. A breast lesion or aberrant echogenic area was scanned in different projections, and the size was determined at its greatest dimension and smallest dimension, respectively. Representative images were saved in the informatics database of Picture Archiving and Communications System.

## Histopathological assessments

The final pathology of the breast lesion and its grade were recorded on the basis of biopsy and/or post-surgical evaluations, which was assessed by two senior pathologists-in-breast. Standard immunohistochemical methods were used to determine the positivity of the tumour for estrogen receptor (ER), progesterone receptor (PR), human epidermal growth factor receptor 2 (HER2), and Ki-67 protein level.

## Statistical analysis

Continuous variables with normal distribution are represented as mean ± standard deviation, while those without the normal distribution are represented as median (interquartile range). Categorical classified variables are represented as numbers (percentage). Primary descriptive statistics of the study were reported. Incidence of malignant NMLs was calculated. Because the irregular indetermined shape, ambiguous contour and blurring margin, the size of breast NMLs was difficult to measure, and therefore it was not calculated. Odds ratio (OR) was determined for the variable analysis. Independent sample t-test was used for the analyses of continuous variables. The receiver operating characteristic (ROC) curve was drawn, the area under the ROC curve (AUC) was calculated, and Youden index was determined and taken as the

**Table 2. Distribution of different breast non-mass lesions.**

| Pathology | | Number(%) |
|---|---|---|
| **Benign lesion** | | |
| | Breast adenosis | 4 (33.33) |
| | Mammary duct ectasia | 1 (8.33) |
| | Fibrocystic breast changes | 1 (8.33) |
| | Granulomatous lobular mastitis | 3 (25.00) |
| | Usual ductal hyperplasia | 1 (8.33) |
| | Usual ductal ectasia with | 1 (8.33) |
| | apocrine metaplasia | |
| | Aberrant adipose tissues | 1 (8.33) |
| **Total** | | 12(100.00) |
| **Malignant lesion** | | |
| | Invasive lobular carcinoma | 4 (8.51) |
| | Invasive carcinoma of no special type | 20 (42.55) |
| | Invasive ductal carcinoma | 8 (17.02) |
| | Ductal carcinoma in situs | 13 (27.66) |
| | Intraductal solid papillary carcinoma | 2 (4.26) |
| **Total** | | 47(100.00) |

Note: Unless otherwise indicated, data are numbers of NMLs and numbers in parentheses are percentages.

cut-off value for the determination of malignancy risk stratification of the sensitivity and specificity. The positive predictive value(PPV) and negative predictive value(NPV) were calculated. All analyses were conducted using SPSS software for Windows, version 23 (IBM Corp, Armonk, NY, USA) and/or Medcalc statistical software version 15.2.2 (Medcalc software BVBA, Ostend, Belgium), and a $p < 0.05$ was considered statistically significant.

## Results

The incidence of malignant NMLs was 4.59% (47/1024) of all pathologically confirmed breast malignant lesions. Analyses of baseline characteristics of the 59 NMLs are summarized in **Table 1**. Pathologies of the 59 NMLs are listed in **Table 2**.

The distribution and malignancy rate of different classifications of BI-RADS are summarized in **Table 3**.

Of the 12 benign NMLs, one NML was classified as BI-RADS category 3, four NMLs were classified as BI-RADS 4A, five NMLs were classified as BI-RADS 4B, and two NMLs were

**Table 3. Distributions of breast NMLs and malignancy rates in BI-RADS 3, 4 and 5.**

| BI-RADS category | NMLs | Malignant NMLs and rate | Likelihood of cancer in the BI-RADS (%) |
|---|---|---|---|
| **3** | 4 | 2(4.3) | >0 but≤2 |
| **4A** | 7 | 5(10.6) | >2 but≤10 |
| **4B** | 14 | 9(19.2) | >10 but≤50 |
| **4C** | 14 | 12(25.5) | >50 but <95 |
| **5** | 20 | 19(40.4) | ≥95 |
| **Total** | 59 | 59(100.0) | |

Note: Unless otherwise indicated, data are numbers of NMLs and numbers in parentheses are percentages. NMLs: non-mass lesions.

**Table 4. Distribution of breast NMLs with different ultrasound features.**

| Features on ultrasound image | | Malignant NML (n = 47) | Benign NML (n = 12) | P value |
|---|---|---|---|---|
| **Punctate hyperechoic foci (microcalcifications)** | | | | 0.001 |
| | Yes | 32(68.1) | 2(16.7) | |
| | No | 15(31.9) | 10(83.3) | |
| **Non-ductal hypoechoic area** | | | | 0.775 |
| | Yes | 26(55.3) | 7(58.3) | |
| | No | 21(44.7) | 5(41.7) | |
| **Posterior shadowing** | | | | 0.036 |
| | Yes | 19(40.4) | 1 (8.3) | |
| | No | 28(59.6) | 11 (91.7) | |
| **Abnormal ductal change** | | | | |
| | Yes | 1(2.1) | 1(8.3) | 0.868 |
| | No | 46(97.9) | 11(91.7) | |
| **Architectural distortion** | | | | 0.467 |
| | Yes | 2(4.3) | 0(0.0) | |
| | No | 45(95.7) | 12(100.0) | |

Note: Unless otherwise indicated, data are numbers of non-mass lesions and numbers in parentheses are percentages. NMLs: non-mass lesions

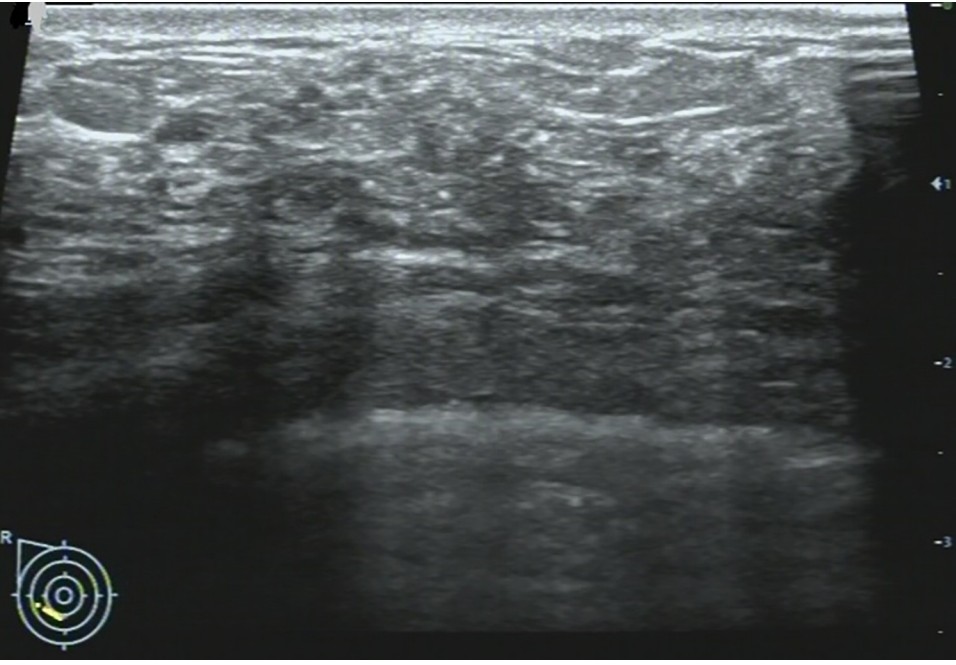

**Fig 2. 36-year-old woman with a non-mass lesion in the right breast.** Image shows the breast lesion locates at outer inferior quadrant, shows irregular heterogeneous isoechoic appearance without space-occupying, with scattering fine hyperechoic foci (microcalcifications) in it, and the border and margin is ambiguous. It is pathologically confirmed a non-special invasive breast carcinoma.

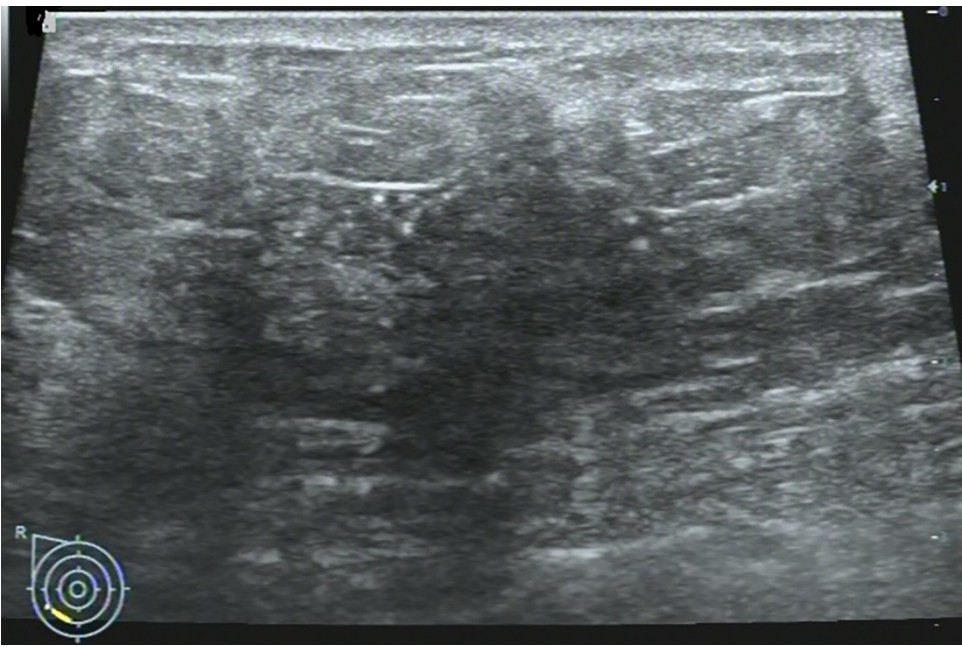

**Fig 3. 67-year-old woman with a non-mass lesion in the right breast.** Image shows the breast lesion locates at outer inferior quadrant, irregular heterogeneous hypoechoic appearance without space-occupying, with scattering fine hyperechoic foci (microcalcifications) and posterior shadowing, and the border and margin is ambiguous. It is pathologically confirmed a breast ductal carcinoma in situs.

classified as BI-RADS 4C. The malignancy rate of NMLs was not concordant with the likelihood of malignancy in the BI-RADS. The NMLs classified as BI-RADS category 3 and category 4A were higher than the corresponding likelihood of malignancy in the BI-RADS, while the category 5 was substantially lower than the corresponding likelihood of malignancy in the BI-RADS. Of the 59 breast NMLs, the main US features are summarized in **Table 4** and illustrated in **Figs 2–6**.

There were significant differences between malignant and benign NMLs for punctate hyperechoic foci and posterior shadowing (all $p<0.05$). If BI-RADS category 4B was taken as a cutoff value, the sensitivity, specificity, AUC, PPV, NPV, and OR were 82.98%(69.19%-92.35%), 41.67%(15.17%-72.33%), 0.62(0.49–0.75), 84.78%(71.13%-93.66%), 38.46%(13.86%-68.42%) and 3.48, respectively.

## Discussion

On MRI NML refers to an enhancing area without an associated mass in shape in the BI-RADS lexicon, whose internal enhancement characteristics can be distinguished from the normal surrounding breast parenchymal enhancement, and the enhancement is distinct [1, 2]. NML on US is different from that found on MRI, apart for some associated findings of tiny cysts, dilated duct, some calcifications, posterior shadowing, nothing of the lesion is distinct [6]. Our findings were consistent with the above summary, as shown on **Figs 2–6**.

Previous studies showed that there was a high malignancy risk in breast NMLs [2, 7–10]. A study by Park et al. showed that 46.2% (330/715) of breast NMLs were malignant, and 53.8% (385/715) of breast NMLs were benign [2]. Kim et al. reported that the malignant breast NMLs were found in 16% (30/186) of 186 pathologically confirmed NMLs, the benign NMLs were in 84% (156/186), and the incidence of NMLs was 5.3% (505/9528) [7]. In a study by Park et al.,

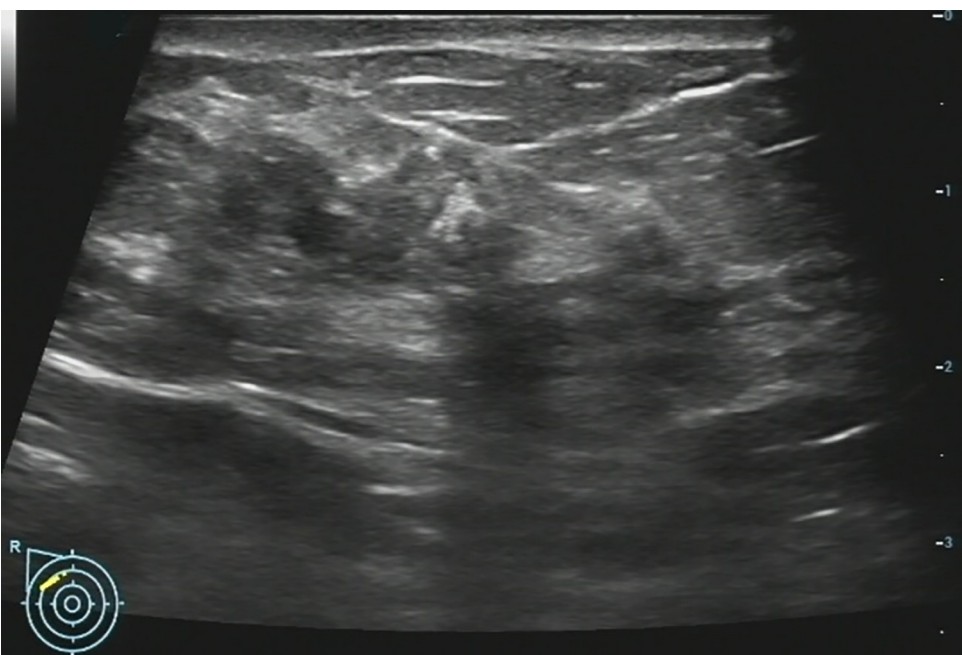

**Fig 4. A 47-year-old woman with a non-mass lesion in the right breast.** Image shows the breast lesion locates at outer inferior quadrant, irregular heterogeneous hypoechoic appearance without space-occupying, with scattering fine hyperechoic foci (microcalcifications) and posterior shadowing, and the border and margin is ambiguous. It is pathologically confirmed a breast adenosis with discrete adenoma-like proliferation.

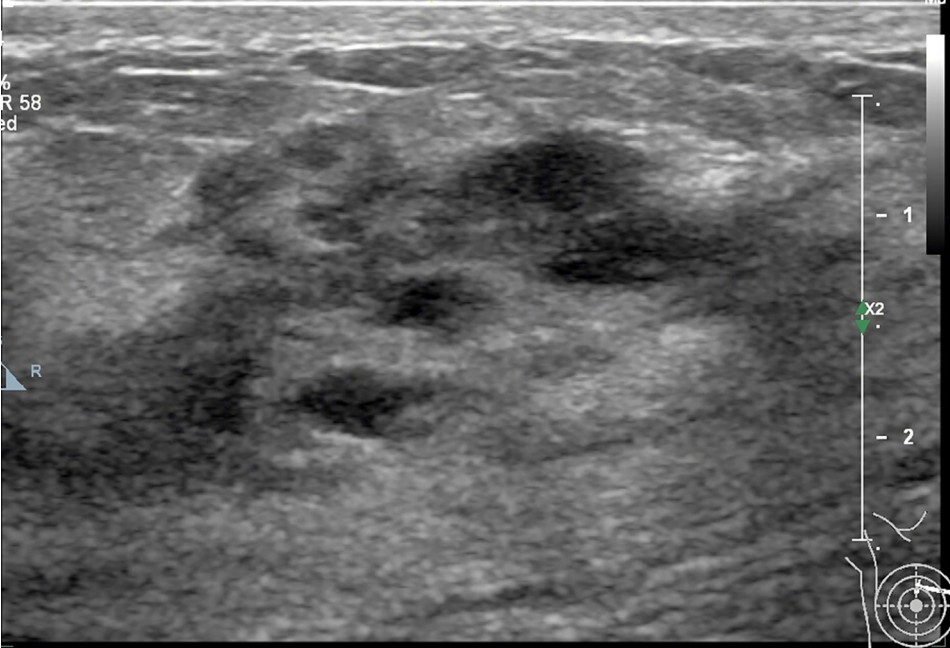

**Fig 5. 47-year-old woman with a non-mass lesion in the right breast.** Image shows the breast lesion locates at internal superior quadrant, markedly hypoechoic regions, and the border and margin is ambiguous. It is pathologically confirmed a breast ductal carcinoma in situs.

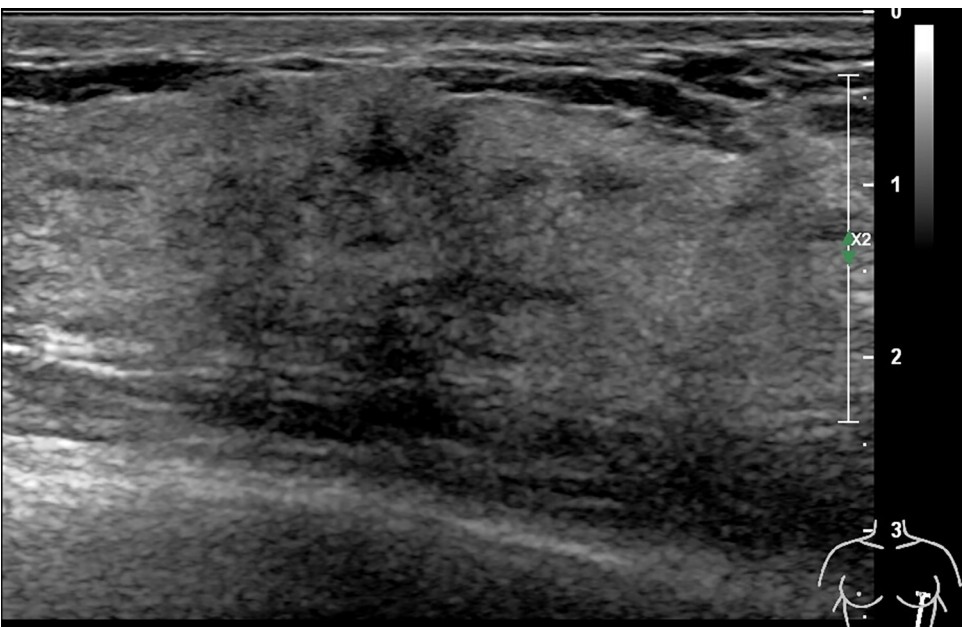

**Fig 6. 32-year-old with a non-mass lesion in the right breast.** Image shows the breast lesion locates at internal inferior quadrant, irregular heterogeneous isoechoic appearance without space-occupying, with scattering fine hyperechoic foci (microcalcifications) in it, and the border and margin is indiscernible. It is pathologically confirmed a focal adipose tissue.

the results showed that 27.3%(33/121) of NMLs were malignant, and 72.7% (88/121) of NMLs were benign [8]. Based on a large sample of screening breast US, Lee et al. founded that the likelihood of malignancy of the NMLs was 2.2% (2/93), the PPV of biopsy was 6.3%, that 95 NMLs were detected in 88 patients from a total of 17,868 screening breast US in 8,856 asymptomatic patients, and the incidence of the NML was 1% [9]. Wang et al. reported that, in their study, malignant NMLs of 53.8% (43/80) were more than benign NMLs of 46.2% (37/80) [10]. In our study, the total number of breast NMLs were not available, so the incidences of breast NMLs and overall malignant NMLs were not calculated. However, we observed the incidence of 4.59% (47/1024) of malignant NMLs from all pathologically confirmed malignant breast lesions. Considering the relatively low probability of NMLs in all breast lesions according to other studies, the incidence of malignant NMLs in all breast NMLs in our study would be far more than 4.59%, so more attention should be paid to breast NMLs.

Ultrasound imaging exhibited promising performance in the malignancy risk stratification of breast masses. A study by Stavros et al. showed that a sensitivity of 98.4% and a NPV of 99.5% were obtained for the assessment of 750 breast solid nodules (625 benign nodules, 125 malignant nodules) in 622 women [5]. In another study by eight experienced radiologists from two countries, they found that PPVs increased with BI-RADS category based on US BI-RADS version 1. The PPVs pre- and post-guidelines for category 3 were 6.0% and 4.4%, for category 4a were 27.3% and 30.5%, for category 4b were 49.9% and 51.5%, for category 4c were 69.0% and 67.4%, and for category 5 were 79.3% and 80.1% [11]. US BI-RADS for the malignancy risk evaluation of breast masses of BIRADS 3–5 lesions showed promising performance, with sensitivity of 92%, specificity of 85% and accuracy of 87% [4]. However, stratifying malignancy risk of breast NMLs using BI-RADS cannot get satisfactory performance. In our study, the sensitivity, specificity, and NPV were 82.98%, 41.67% and 38.46%, respectively, which were lower than the study by Wang et al. [10] that the sensitivity of 95.35%, specificity of 43.24%, and

NPV of 88.89% of US for NMLs; and the PPV of 84.78% in our study was higher than the PPV of 66.13% obtained by Wang et al. [10]. The AUC of 0.62 in our study was lower than the accuracy of 71.25% by Wang et al. [10] and the AUC of 0.908–0.911 by Park et al. [3], the reason may be that the incidences of these US features are various in different study population, which affect the malignancy risk stratification in different degree. The lower specificity indicates that US has problem of higher false positive rate for malignancy risk stratification of breast NMLs.

Previous study has identified the descriptors that nonparallel orientation of the breast nodule, spiculated margins, angular margins, microcalcifications, or posterior shadowing that are commonly associated with histopathologically malignant nodules [5]. However, except for the likelihood of malignancy, there is little information regarding which descriptors are most strongly associated with malignancy in nodules of BI-RADS 3 and 4 as well as to what extent a single descriptor should be considered for the final evaluation of malignancy risk; and there are no definite US features for category 4 and 5 in the BI-RADS [5, 6]. In regard to the assessment of NMLs, descriptors of spiculated and angular margins seem applicable or ambiguous for majority of the NMLs, circumscribed and well-defined margins, round and ovoid shapes are not applicable for almost all NMLs, and only the rest descriptors are applicable; these lead to a compromise for the proper BI-RADS classification. In this study, many NMLs were classified as 4A, which suggests >2% to≤10% likelihood of malignancy according to the ACR recommendation of BI-RADS [6]. Given BI-RADS 4B as the cutoff value, many NMLs with BI-RADS 4A were regarded as benign lesions. There are suggestive features for category 3 in the BI-RADS (a solid mass with a circumscribed margin, oval shape, and parallel orientation, and an isolated complicated cyst) [6], but the category may be adjusted by upgrading and downgrading, therefore, some NMLs without the typical features of BI-RADS category 3 in this study were classified as category 4A and overestimated for malignancy. On the other hand, some malignant NMLs were classified into a lower BI-RADS category and underestimated for malignancy. As a result, the PPVs were still excellent, but the NPVs were very poor (38.46%). The malignancy rates of NMLs were not concordant with the likelihood of malignancy in the BI-RADS, the malignancy rates of NMLs classified as BI-RADS category 3 (6.4% and 4.3%) and category 4A (all 10.6%) were higher than the corresponding likelihood of malignancy in the recommended BI-RADS. However, the malignancy rates of NMLs classified as BI-RADS category 4C and 5 were from 25.5% to 40.4%, which were substantial lower than the likelihood of malignancy of >50% to ≥95% in the recommended BI-RADS [6].

Microcalcifications associate dominantly with breast cancer in breast NMLs [2, 3, 7, 8]. In our study, 68.1% (32/47) of malignant NMLs showed microcalcifications, as did 16.7% (2/12) of benign NMLs. These were different from the previous studies [3, 7, 8]. Park et al. [3] reported that 72.3 (154/213) of malignant NMLs showed calcifications, whereas 10.1% (25/247) of benign NMLs showed calcifications; Kim et al. [7] reported that calcifications appeared in 40% (12/30) of malignant NMLs, whereas in the benign NMLs there were no visible calcifications; Park et al. [8] reported that 27.3% (9/33) of malignant NMLs showed calcifications, while only 10.2% (9/88) of benign NMLs did. The reasons for various incidences of microcalcifications may be that they usually present as punctate hyperechoic foci on US images (and, while not all punctate hyperechoic foci are microcalcifications, some microcalcifications do not present hyperechoic); in addition, the confounder of different study population; the presence of inter-observer agreement; and the variant of different US systems. Mammography is superior to US for detecting microcalcifications and remains the gold standard [3, 12]. In a study, Grigoryev et al. reported that the sensitivity and specificity for detecting microcalcifications were 70% and 30% in US and 45% and 55% in mammography, respectively; and the PPV of 88% for mammography was superior to that of 78% for US [12].

Posterior shadowing often appears following scar tissue, macrocalcifications, and clustered microcalcifications, which highly associate with breast carcinoma [2, 3, 13]. In this study, posterior shadowing appeared in 8.3% of benign NMLs ($P$ = 0.036), while it appeared in 40.4% (19/47) of malignant NMLs. This is a higher incidence than that of 26.8% (57/156) of malignant breast NMLs reported by Park et al. [3].

In our study, non-ductal hypoechoic area presented in 55.3% (26/47) of malignant NMLs and 41.7% (2/12) of benign NMLs, with or without microcalcifications. This was higher than in the report by Wang et al., in which 92.5% (74/80) of breast NMLs presented as a hypoechoic area and a hypoechoic area with microcalcification [10]. These results showed that non-ductal hypoechoic area is the main common feature of malignant and benign NMLs.

In this study, 4.3% (2/47) of malignant NMLs were found with architectural distortion, and all benign NMLs without. Park et al. reported that 6.6% (14/213) of malignant NMLs and 4.0% (10/247) of benign NMLs had architectural distortion on US images [3]. Wang et al. reported that in 80 breast NMLs, 5% had architectural distortion [10].

A dilated ductus without a tumor in it is usually associated with benign breast diseases. In this study, abnormal ductal changes, such as a tortuous dilated ductus, were seen in 2.1% (1/47) of malignant and 8.3% (1/12) of benign NMLs. In a study by Wang et al., there were two visible ducts with solid entities in 80 NMLs (2.5%; 2/80) [10]. These suggest that abnormal ductal change is not common in malignant NMLs.

The size of breast carcinomas is an important variable for clinical management, metastasis, and prognosis, and inadequate measurement may be misleading [14]. Because taking an accurate measurement in breast NMLs is difficult, no size measurement was performed in our study.

In future, ultrasound elastography and/ or contrast-enhanced ultrasound may combined with conventional US to improve assessment of breast non-mass lesions [15–17]; new descriptors may be adopted and/or incorporated with BI-RADS for the malignancy risk stratification of breast masses and NMLs, and the stratification efficacy may be improvable [18].

Collectively, it is hard to distinguish malignant from benign NMLs by non-ductal hypoechoic areas and dilated ductus on US, and punctate hyperechoic foci indicating microcalcifications and posterior shadowing are important features significantly suggestive of malignant breast NMLs. Stratifying the malignancy risk of breast NMLs using BI-RADS in a protocol analogous to that the BI-RADS for breast masse lesions cannot obtain satisfactory results except the sensitivity and PPV, and it may compromise the overall stratification efficacy. To identify and separate NMLs from breast masses and perform different assessment protocols will be a rational solution.

The strengths of this study are that the breast NMLs were derived from a histopathologically confirmed large sample, and the incidence of breast malignant NMLs and other associated variables calculated are accurate, reliable and robust.

There was limitation in this study. The source sample was large, but the sample for NMLs study was small, and many benign NMLs without histological results were excluded, which may cause sample selection bias.

## Conclusions

Breast NMLs present various US appearances, breast carcinoma is not rare in the NMLs, and the incidence of malignant NMLs was 4.59% of all breast carcinomas. Non-ductal hypoechogenicity, microcalcifications and posterior shadowing are common findings in the breast NMLs, and microcalcifications and posterior shadowing associate significantly with breast carcinoma. In stratifying the malignancy risk of breast NMLs using BI-RADS in a protocol

analogous to that the BI-RADS for breast masse lesions, some benign NMLs are upgraded in category, and their likelihood of malignancy is overestimated, some malignant NMLs are underestimated in their likelihood of malignancy. If BI-RADS 4B is taken as a cutoff value, the sensitivity and PPV for the malignancy risk is promising, but the specificity and NPV are very poor. The solution may be that to identify and separate NMLs from breast masses and use different malignancy risk stratification protocols (S1 and S2 Data).

## Supporting information

**S1 Data. Finally enrolled patients.** Table of number of subjects, histopathology, BI-RADS classifications, and locations of breast non-mass lesions of the finally enrolled patients. (XLSX)

**S2 Data. Figures of breast non-mass lesions.** Representative sonographic images of breast non-mass lesions. (ZIP)

## Author Contributions

**Conceptualization:** Size Wu.

**Data curation:** Mingnan Lin, Size Wu.

**Formal analysis:** Mingnan Lin, Size Wu.

**Funding acquisition:** Size Wu.

**Investigation:** Mingnan Lin.

**Methodology:** Size Wu.

**Supervision:** Size Wu.

**Validation:** Size Wu.

**Writing – original draft:** Mingnan Lin.

**Writing – review & editing:** Size Wu.

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
