## [Decision Letter · Decision Letter 0]

22 Sep 2022

PONE-D-22-24181Ultrasound classification of non-mass breast lesions following BI-RADS presents high positive predictive valuePLOS ONE

Dear Dr. Wu,

Thank you for submitting your manuscript to PLOS ONE. After careful consideration, we feel that it has merit but does not fully meet PLOS ONE’s publication criteria as it currently stands. Therefore, we invite you to submit a revised version of the manuscript that addresses the points raised during the review process.

We look forward to receiving your revised manuscript.

Kind regards,

Ibrahim Umar Garzali, MBBS, FWACS

Academic Editor

PLOS ONE

**2. **In your Data Availability statement, you have not specified where the minimal data set underlying the results described in your manuscript can be found. PLOS defines a study's minimal data set as the underlying data used to reach the conclusions drawn in the manuscript and any additional data required to replicate the reported study findings in their entirety. All PLOS journals require that the minimal data set be made fully available. For more information about our data policy, please see http://journals.plos.org/plosone/s/data-availability.

Reviewers' comments:

Reviewer's Responses to Questions

**Comments to the Author**

1. Is the manuscript technically sound, and do the data support the conclusions?

Reviewer #1: Yes

Reviewer #2: No

2. Has the statistical analysis been performed appropriately and rigorously? 

Reviewer #1: Yes

Reviewer #2: No

3. Have the authors made all data underlying the findings in their manuscript fully available?

Reviewer #1: Yes

Reviewer #2: Yes

4. Is the manuscript presented in an intelligible fashion and written in standard English?

Reviewer #1: Yes

Reviewer #2: Yes

5. Review Comments to the Author

Reviewer #1: i read the article with an interest. non mass lesions is an important and challenging issue in breast cancer. the authors give the satisfying results for this entity. thanks for all authors for this study.

Reviewer #2: The title and objectives are different. the title talks about positive predictive value while the objective is mainly on incident of malignancy in NML by USS. The methodology contains some results like age and bio-demographic characteristics.

odd ratio should not have calculated because the study is not trying to ascertain cause or risk factors for breast cancer. The figures reflecting USS findings suggestive of mass lesions should not been part of the result. for the fact that the study is a retrospective one, sensitivity and predictive value should not have considered because standardization could not be made as such the sensitivity would be very low.

Discussion was initially off point as the study was on uss findings and not MRI. Most of the analyses were not in keeping with objective of the study. The discussion was mainly analysing microcalcification, punctate hypoechoic foci and posterior acoustic shadow.

The author did well by stating the AUC of 0.62 was low compared to other stuydy.

the conclusion ought to have qoute the low incidence of malignancy in all patients with NML not just NML with histological confirmation compared to all maligmant breast lesions.

The references are very scanty. authors should have qouted copiously from the literature especially during the intrioduction and discussion.

the sensitivity and specificity are low and not promising as stated and therefore the conclusion should have reflected this reality since p<0.05 was regarded as statistically significant

6. PLOS authors have the option to publish the peer review history of their article (what does this mean?). If published, this will include your full peer review and any attached files.

Reviewer #1: **Yes: **Kemal EYVAZ

Reviewer #2: No

---

## [Author Response · Author response to Decision Letter 0]

20 Oct 2022

Response Letter

Dear Editor, 

On behalf of all authors, I would like to resubmitted the revised manuscript (PONE-D-22-24181) entitled Ultrasound classification of non-mass breast lesions following BI-RADS presents high positive predictive value. The manuscript has been revised following the reviewers’ comments, and detailed response was given trackable in the revised manuscript. I believe that the revised manuscript may meet the criteria of the prestigious journal of PLOS ONE for publication in general. Please feel free to contact me (as the corresponding author) should further clarity be needed. I’m looking forward to hearing from you.

Sincerely,

Prof. Size Wu

The corresponding author

Responses to the Reviewer's Questions

Comments: 1. Is the manuscript technically sound, and do the data support the conclusions?

Reviewer #1: Yes

Responses: Thanks.

Reviewer #2: No

Responses: The aims of our manuscript were to investigate the performance (diagnostic efficacy) of stratifying the malignancy risk of non-mass breast lesions (NMLs) following breast imaging reporting and data system (BI-RADS), including sensitivity, specificity, area under the receiver operating characteristic curve (AUC) (accuracy), positive and negative predictive values. Among them, positive predictive value is a useful parameter for the interpretation of the role of BI-RADS for NMLs. There are numerous publishing relating to the assessment of breast masses, however, the studies of US for the assessment of breast non-masses are much fewer. In fact, NMLs were not rare, they were frequently overlooked in clinical evaluation, for some physicians regarded them as atypical breast masses. On breast X ray and sonography, some breast mass and NMLs showed similar appearance, on contrast-enhanced MRI, breast mass and NMLs presented different features, and thus the conception and terminology of NML were coined. On ultrasound images, typical breast mass was with distinct appearance, even in small size. While on X ray some typical small size breast mass presented ambiguous contour, and less attention had paid for them in past time. In previous routine clinical practice, expect some case of NMLs were categorized as BI-RADS 0, other cases were actually classified as a certain category (breast mass). Strictly speaking, classifying breast NMLs using BI-RADS is not appropriated. We believe that confusion of breast mass and NML may result in an incomplete and not sound evaluation. In this study, a large sample of population were reviewed to determine the incidence of NMLs, to better understand the NMLs. Majority of NMLs were benign lesions, did not undergo biopsy or operation, and absence of histological results, so it’s hard to determine accurately, while malignant NMLs were undergone biopsy or surgical treatment, with histological results available, thus the diagnosis was sound and vigorous. So we investigated the incidence of malignant NMLs in this study, and the result was sound and vigorous. For the process of data, criteria of inclusion and exclusion were followed strictly, with appropriate design, and the statistical analyses were competent. The protocols of ultrasound imaging for the breast were following standard specification, as descripted in the Methods section (Acquisition of US images of the breast lesions). The conclusions drawn were appropriately.

Comments: 2. Has the statistical analysis been performed appropriately and rigorously?

Reviewer #1: Yes

Responses: Thanks.

Reviewer #2: No

Responses: For the process of data, an appropriate design was conducted, the statistical analyses were performed appropriately, and also with consultation to expertise in statistics.

Comments: 3. Have the authors made all data underlying the findings in their manuscript fully available?

Reviewer #1: Yes

Responses: Thanks. 

Reviewer #2: Yes

Responses: Thanks.

Comments: 4. Is the manuscript presented in an intelligible fashion and written in standard English?

Reviewer #1: Yes

Responses: Thanks.

Reviewer #2: Yes

Responses: Thanks.

Comments: Reviewer #1: i read the article with an interest. non mass lesions is an important and challenging issue in breast cancer. the authors give the satisfying results for this entity. thanks for all authors for this study.

Responses: Thanks.

Comments: Reviewer #2: The title and objectives are different. the title talks about positive predictive value while the objective is mainly on incident of malignancy in NML by US. 

Responses: Agreed. The “objective” was revised. The aims of our manuscript were to investigate the performance (diagnostic efficacy) of stratifying the malignancy risk of non-mass breast lesions (NMLs) following breast imaging reporting and data system (BI-RADS), including sensitivity, specificity, area under the receiver operating characteristic curve (AUC) (accuracy), positive and negative predictive values. Among them, positive predictive value is a useful parameter for the interpretation of the role of BI-RADS for NMLs, to use it as a title can highlight the main finding of this study. Of course, there were other findings that using the BI-RADS for the assessment, the likelihood of malignancy of malignant NMLs is underestimated, and that of benign NMLs is overestimated, these are challenge and drawbacks.

To better understand the NMLs, incident of malignancy was analyzed. Majority of NMLs were benign lesions, did not undergo biopsy or operation, and absence of histological results, so it’s hard to determine accurately, while malignant NMLs were undergone biopsy or surgical treatment, with histological results available, thus the diagnosis was sound and reliable. So we investigated the incidence of malignant NMLs in this study, and the result was sound and reliable.

Comments: The methodology contains some results like age and bio-demographic characteristics. odd ratio should not have calculated because the study is not trying to ascertain cause or risk factors for breast cancer. 

Responses: ## To save page and shorted the length of the paper, age and bio-demographic characteristics were incorporated into a table. Similar format style can be found in other article, e.g., (1) Kim et al. Radiology. 2018;288(1):55-63. doi: 10.1148/radiol.2018171987. (2) Zhuang et al. Radiology. 2017;283(3):873-882. doi: 10.1148/radiol.2016160131”. Of course, it had better be apart. 

## “odd ratio” has different usages in biomedical study, similar application can be found in articles: (1) Kim et al. Preoperative Axillary US in Early-Stage Breast Cancer: Potential to Prevent Unnecessary Axillary Lymph Node Dissection. Radiology. 2018;288(1):55-63. doi: 10.1148/radiol.2018171987. (2) Stavros, et al. Solid breast nodules: use of sonography to distinguish between benign and malignant lesions. Radiology. 1995;196(1):123-34. doi: 10.1148/radiology.

Comments: The figures reflecting USS findings suggestive of mass lesions should not been part of the result. for the fact that the study is a retrospective one, sensitivity and predictive value should not have considered because standardization could not be made as such the sensitivity would be very low.

Responses: It’s right that the sensitivity and predictive value are used mainly in a prospective study. On the other hand, in a retrospective study, if the design is adequate, the reference standardization is objective and reliable (e.g., histopathological result), the data were collected adequately, then the sensitivity and predictive value can be calculated properly. E.g., Buda et al. Management of Thyroid Nodules Seen on US Images: Deep Learning May Match Performance of Radiologists. Radiology. 2019;292(3):695-701. doi: 10.1148/radiol.2019181343.

Comments: Discussion was initially off point as the study was on USS findings and not MRI. Most of the analyses were not in keeping with objective of the study. 

Responses: Somewhat agreed. It’s right that the purpose of this study was to discuss USS findings, but to avoid abruption of definition of NML, background knowledge was introduced. On breast X ray and sonography, some breast mass and NMLs showed similar appearance, on contrast-enhanced MRI, breast mass and NMLs presented different features, and thus the conception and terminology of NML were coined. The following words were keeping with the objective of the study and associated findings.

Comments: The discussion was mainly analysing microcalcification, punctate hypoechoic foci and posterior acoustic shadow.

Responses: Somewhat agreed. The sensitivity, specificity, AUC, PPV and NPV were dominantly with microcalcification, punctate hypoechoic foci and posterior acoustic shadow in NMLs, so these features were discussed intensively. 

Comments: The author did well by stating the AUC of 0.62 was low compared to other study. the conclusion ought to have qoute the low incidence of malignancy in all patients with NML not just NML with histological confirmation compared to all malignant breast lesions.

Responses: Somewhat agreed. To obtain sound and reliable results and conclusion, we did not use breast lesions without histological results, so many benign lesions were excluded, the results were obtained mainly on malignant lesions, and these were the limitations of this study.

Comments: The references are very scanty. authors should have qouted copiously from the literature especially during the introduction and discussion.

Responses: Agreed. The literature of contrast-enhanced MRI for the assessment of NMLs were rich, but it’s scanty for ultrasound, so there were not enough valuable references for this manuscript. Some references relating to breast mass and US BI-RADS were added, and associated revision was done.

Comments: the sensitivity and specificity are low and not promising as stated and therefore the conclusion should have reflected this reality since p<0.05 was regarded as statistically significant.

Responses: In this study we believed that the present statement may replace other expression, and more rephrase may be omitted.

---

## [Editor Report · Decision Letter 1]

15 Nov 2022

Ultrasound classification of non-mass breast lesions following BI-RADS presents high positive predictive value

PONE-D-22-24181R1

Dear Dr. Wu,

We’re pleased to inform you that your manuscript has been judged scientifically suitable for publication and will be formally accepted for publication once it meets all outstanding technical requirements.

Kind regards,

Ibrahim Umar Garzali, MBBS, FWACS

Academic Editor

PLOS ONE
---

## [Editor Report · Acceptance letter]

18 Nov 2022

PONE-D-22-24181R1 

Ultrasound classification of non-mass breast lesions following BI-RADS presents high positive predictive value 

Dear Dr. Wu:

I'm pleased to inform you that your manuscript has been deemed suitable for publication in PLOS ONE. Congratulations! Your manuscript is now with our production department. 

Kind regards, 

on behalf of

Dr. Ibrahim Umar Garzali 

Academic Editor

PLOS ONE